# Safe Active Learning for Time-Series Modeling with Gaussian Processes

**Christoph Zimmer**    **Mona Meister**    **Duy Nguyen-Tuong**
Bosch Center for Artificial Intelligence, Renningen, Germany
{christoph.zimmer,mona.meister,duy.nguyen-tuong}@de.bosch.com

## Abstract

Learning time-series models is useful for many applications, such as simulation and forecasting. In this study, we consider the problem of actively learning time-series models while taking given safety constraints into account. For time-series modeling we employ a Gaussian process with a nonlinear exogenous input structure. The proposed approach generates data appropriate for time series model learning, i.e. input and output trajectories, by dynamically exploring the input space. The approach parametrizes the input trajectory as consecutive trajectory sections, which are determined stepwise given safety requirements and past observations. We analyze the proposed algorithm and evaluate it empirically on a technical application. The results show the effectiveness of our approach in a realistic technical use case.

## 1 Introduction

Active model learning deals with the problem of sequential data labeling for learning an unknown function. Data points are sequentially selected for labeling such that the information required for approximating the unknown function is maximized, according to some measures. The overall goal is to create an accurate model without having to supply more data than necessary and, thereby reducing the annotation effort and measurement costs. Active learning has been well studied for classification tasks, e.g. for image labeling [12], but in the field of regression, the active learning approach, related to the optimal experimental design problem [8], is not yet widespread.

For actively learning time-series models representing physical systems, the data has to be generated such that the relevant dynamics can be captured. In practice, the physical system needs to be excited by dynamically moving around in the input space using *input trajectories*, such that the collected data, i.e. input and output trajectories, contain as much information about the dynamics as possible. Commonly used input trajectories include sinusoidal functions, ramps and step functions, white noise, etc. [13, 17]. When employing input excitation on physical systems, however, additional aspects of safety need to be considered. The excitation must not damage the physical system while dynamically exploring the input space, making it crucial to identify safe regions where dynamic excitation can be performed.

In this paper we consider the problem of safe exploration for active learning of time-series models. The goal is to generate input trajectories and output measurements which are informative for learning time-series models. To do so, our input trajectories are parametrized in consecutive sections, e.g. as consecutive piecewise ramps or splines. These consecutive sections of the input trajectory are determined *stepwise* in an explorative approach. Given observations, the next trajectory sections are determined by maximizing an information gain criterion with respect to the model. In this paper, we employ a Gaussian process with a *nonlinear exogenous* structure as the time-series model for which an appropriate exploration criterion is desired. An additional Gaussian process model is simultaneously used for predicting safe input regions, given safety requirements. The sections of

the input trajectory are determined by solving a constraint optimization problem, taking the safety prediction into account. The main contributions of the paper can be summarized as:

- We formulate an active learning setting for learning time-series models with dynamic exploration, in the context of the Gaussian process framework.
- We incorporate the safety aspect into the exploration mechanism and derive a criterion appropriate for the dynamic exploration of the input space with trajectories.
- We provide a theoretical analysis of the algorithm, and empirically evaluate the proposed approach on a realistic technical use case.

The remainder of the paper is organized as follows. In Section 2, we provide an overview on related work. In Section 3, we introduce the algorithm for safe active learning of time-series models. Section 4 provides a theoretical analysis, and in Section 5, we highlight our empirical evaluations in learning time-series model in several settings. The Appendix contains the proofs of the theoretical analysis section and some more experimental investigations.

## 2  Related Work

Most existing work for safe exploration in unknown environments is in the reinforcement learning setting [16, 10, 9]. For example, the safe exploration in finite MDP relies on the restriction of suitable policies, ensuring ergodicity at a user-defined safety level [16]. In [10], the ergodic assumption for the MDPs is dropped by introducing fatal absorbing states. In [9], the authors consider the use of a multi-armed, risk-aware bandit setting to prevent hazards when exploring different tasks. Strategies for exploring unknown environments have also been reflected in the framework of global optimization with Gaussian processes [1, 23, 11]. For example, [11] propose an efficient submodular exploration criterion for near-optimal sensor placements, i.e. for discrete input spaces. In [1], a framework is presented which yields a compromise between exploration and exploitation through confidence bounds. In [23], the authors show that under reasonable assumptions, strong exploration guarantees can be given for Bayesian optimization with Gaussian processes.

Safe exploration using Gaussian processes (GP) has also been considered in the past, such as for safe active learning [20] and safe Bayesian optimization [2, 24]. In [24], for example, a two-steps process is proposed for a safe exploration and efficient exploitation of identified safe areas. In safe active learning, [20] proposes a method for safe exploration based on the GP variance for stationary, i.e. pointwise, measurements. In contrast to the work by [20], we consider the setting of safe dynamical exploration, i.e. using trajectory-wise measurements. This setting is especially useful when actively learning time-series models.

The problem of active learning for time-series models has not yet been considered extensively in the machine learning literature. Work on related topics is mostly in the field of online design of experiments, e.g. [6]. In [6], the authors employ a parametric model for learning dynamical processes, in which the data for model learning is obtained by exploring using the Fisher information matrix. In contrast to their work, we explore unknown environments by employing a criterion defined for the non-parametric GP model, while also taking into account safety requirements. Furthermore, our proposed exploration scheme is rigorously analyzed providing further algorithmic insights.

## 3  Safe Active Learning for Time-Series Modeling

Our goal is to approximate an unknown function $f : X \subset \mathbb{R}^d \to Y \subset \mathbb{R}$. In the case of time-series models, e.g. the well-established *nonlinear exogenous* (NX) model, the input space consists of discretized values of the so-called manipulated variables [3]. Thus, $\boldsymbol{x}_k$ at time $k$ can be given as

$$\boldsymbol{x}_k = (\boldsymbol{u}_k, \boldsymbol{u}_{k-1}, \ldots, \boldsymbol{u}_{k-d_2+1}),$$

where $(\boldsymbol{u}_k)_k$, $\boldsymbol{u}_k \in \mathbb{R}^{d_2}$, represents the discretized manipulated trajectory. Here, $d_1$ is the dimension of the system's input space, $d_2$ the dimension of the NX structure, and $d = d_2 \cdot d_1$. In practice, the elements $\boldsymbol{u}_k$ are measured from physical systems and need not be equidistant, however, for notational convenience we assume equidistance in this setting. In general, the manipulated trajectories are continuous signals and can be explicitly controlled. In the model learning setting, we observe data in

the form of $n$ consecutive piecewise trajectories $\mathcal{D}_n^f = \{\boldsymbol{\tau}_i, \boldsymbol{\rho}_i\}_{i=1}^n$, where the input trajectory $\boldsymbol{\tau}_i$ is a matrix and consists of $m$ input points of dimension $d$, i.e. $\boldsymbol{\tau}_i = (\boldsymbol{x}_1^i, \ldots, \boldsymbol{x}_m^i) \in \mathbb{R}^{d \times m}$. The output trajectory $\boldsymbol{\rho}_i$ contains $m$ corresponding output measurements, i.e. $\boldsymbol{\rho}_i = (y_1^i, \ldots, y_m^i) \in \mathbb{R}^m$.

The considered problem is to determine the next piecewise trajectory $\boldsymbol{\tau}_{n+1}$ as input excitation to the physical system such that the *information gain* of $\mathcal{D}_{n+1}^f$ – with respect to modeling $f$ – is increased. At the same time, $\boldsymbol{\tau}_{n+1}$ should be determined subject to given safety constraints. In this section we elaborate on the setting and describe the algorithm. The definition of the considered information gain and corresponding analysis are provided in Section 4.

## 3.1 Modeling Trajectories with Gaussian Processes

We employ a Gaussian Process (GP) model to approximate the function $f$ (see [19] for more details). A GP is specified by its mean function $\mu(\boldsymbol{x})$ and covariance function $k(\boldsymbol{x}_i, \boldsymbol{x}_j)$, i.e. $f(\boldsymbol{x}_i) \sim \mathcal{GP}(\mu(\boldsymbol{x}_i), k(\boldsymbol{x}_i, \boldsymbol{x}_j))$. Given noisy observations of input and output trajectories, the joint distribution according to the GP prior is given as

$$p(\boldsymbol{P}_n | \boldsymbol{T}_n) = \mathcal{N}\left(\boldsymbol{P}_n | \boldsymbol{0}, \boldsymbol{K}_n + \sigma^2 \boldsymbol{I}\right),$$

where $\boldsymbol{P}_n \in \mathbb{R}^{n \cdot m}$ is a vector concatenating output trajectories and $\boldsymbol{T}_n \in \mathbb{R}^{n \cdot m \times d}$ a matrix containing input trajectories. The covariance matrix is represented by $\boldsymbol{K}_n \in \mathbb{R}^{n \cdot m \times n \cdot m}$. In this paper, we employ the Gaussian kernel as the covariance function, i.e. $k(\boldsymbol{x}_i, \boldsymbol{x}_j) = \sigma_f^2 \exp(-\frac{1}{2}(\boldsymbol{x}_i - \boldsymbol{x}_j)^T \boldsymbol{\Lambda}_f^2 (\boldsymbol{x}_i - \boldsymbol{x}_j))$, which is parametrized by $\boldsymbol{\theta}_f = (\sigma_f^2, \boldsymbol{\Lambda}_f^2)$. Furthermore, we have a zero vector $\boldsymbol{0} \in \mathbb{R}^{n \cdot m}$ as mean, an $n \cdot m$-dimensional identity matrix $\boldsymbol{I}$, and $\sigma^2$ as output noise variance (see [19]). Given the joint distribution, the predictive distribution $p(\boldsymbol{\rho}^* | \boldsymbol{\tau}^*, \mathcal{D}_n^f)$ for a new piecewise trajectory $\boldsymbol{\tau}^*$ can be expressed as

$$p(\boldsymbol{\rho}^* | \boldsymbol{\tau}^*, \mathcal{D}_n^f) = \mathcal{N}\left(\boldsymbol{\rho}^* | \boldsymbol{\mu}(\boldsymbol{\tau}^*), \boldsymbol{\Sigma}(\boldsymbol{\tau}^*)\right), \tag{1}$$

with

$$\begin{aligned}
\boldsymbol{\mu}(\boldsymbol{\tau}^*) &= \boldsymbol{k}(\boldsymbol{T}_n, \boldsymbol{\tau}^*)^T (\boldsymbol{K}_n + \sigma^2 \boldsymbol{I})^{-1} \boldsymbol{P}_n, \\
\boldsymbol{\Sigma}(\boldsymbol{\tau}^*) &= \boldsymbol{k}^{**}(\boldsymbol{\tau}^*, \boldsymbol{\tau}^*) - \boldsymbol{k}(\boldsymbol{T}_n, \boldsymbol{\tau}^*)^T (\boldsymbol{K}_n + \sigma^2 \boldsymbol{I})^{-1} \boldsymbol{k}(\boldsymbol{T}_n, \boldsymbol{\tau}^*),
\end{aligned} \tag{2}$$

where $\boldsymbol{k}^{**} \in \mathbb{R}^{m \times m}$ is a matrix with $k_{ij}^{**} = k(\boldsymbol{x}_i, \boldsymbol{x}_j)$. The matrix $\boldsymbol{k} \in \mathbb{R}^{n \cdot m \times m}$ contains kernel evaluations relating $\boldsymbol{\tau}^*$ to the previous $n$ input trajectories. As the covariance matrix $\boldsymbol{K}_n$ is fully occupied, the input points $\boldsymbol{x}$ are fully correlated within a piecewise trajectory, as well as across different trajectories; this enables the exploitation of high capacity correlations. However, due to the potentially large dimension $n \cdot m$, inverting the matrix $\boldsymbol{K}_n + \sigma^2 \boldsymbol{I}$ can be infeasible. GP approximation techniques can be employed, e.g. using sparse inducing inputs or variational approaches [18, 22, 26].

## 3.2 Modeling the Safety Condition

The safety status of the system is described by an unknown function $g : X \subset \mathbb{R}^d \to Z \subset \mathbb{R}$, mapping an input point $\boldsymbol{x}$ to a safety value $z$, which acts as a safety indicator. The values $z$ are computed using information from the system, and are designed such that all values equal or greater than zero are considered safe for the corresponding input $\boldsymbol{x}$. Example 1 shows a construction for computing $z$ values. More examples can be found in the evaluation in Section 5.

**Example 1** (A safety indicator for a high-pressure fluid system). *In a high-pressure fluid system, we can measure the pressure $\psi$ for a given input state $\boldsymbol{x}$. Additionally, we know the value of the maximal pressure $\psi_{max}$ which can act on the physical system. Given the current pressure $\psi$, the safety values $z$ can be computed as*

$$z(\psi) = 1 - exp((\psi - \psi_{max})/\lambda_p), \tag{3}$$

*where $\lambda_p$ describes the decline, when $\psi$ increases towards $\psi_{max}$.*

Note that $z$ is continuous and, intuitively, indicates the distance of a given point $\boldsymbol{x}$ from the *unknown* safety boundary in the input space. Thus, given the function $g$ – or an estimate of it – we can evaluate the level of safety for a trajectory $\boldsymbol{\tau}$. We consider a trajectory as safe, if the probability that its safety values $z$ are greater than zero is sufficiently large, i.e.

$$\int_{z_1, \ldots, z_m \geq 0} p(z_1, \ldots, z_m | \boldsymbol{\tau}) \, dz_1, \ldots, z_m > 1 - \alpha,$$

with $\alpha \in (0, 1]$ representing the threshold for considering $\boldsymbol{\tau}$ unsafe. Given data $\mathcal{D}_n^g = \{\boldsymbol{\tau}_i, \boldsymbol{\zeta}_i\}_{i=1}^n$, with $\boldsymbol{\zeta}_i = (z_1^i, \ldots, z_m^i) \in \mathbb{R}^m$, we employ a GP to approximate the function $g$. The predictive distribution $p(\boldsymbol{\zeta}^* | \boldsymbol{\tau}^*, \mathcal{D}_n^g)$ given a piecewise trajectory $\boldsymbol{\tau}^*$ is then computed as

$$p(\boldsymbol{\zeta}^* | \boldsymbol{\tau}^*, \mathcal{D}_n^g) \ = \ \mathcal{N}\left(\boldsymbol{\zeta}^* | \boldsymbol{\mu}_g(\boldsymbol{\tau}^*), \boldsymbol{\Sigma}_g(\boldsymbol{\tau}^*)\right) , \tag{4}$$

with $\boldsymbol{\mu}_g(\boldsymbol{\tau}^*)$ and $\boldsymbol{\Sigma}_g(\boldsymbol{\tau}^*)$ being the corresponding mean and covariance. The quantities $\boldsymbol{\mu}_g$ and $\boldsymbol{\Sigma}_g$ are computed as shown in Eq. (2), then with $\boldsymbol{Z}_n \in \mathbb{R}^{n \cdot m}$ as the target vector concatenating all $\boldsymbol{\zeta}_i$. By employing a GP for approximating $g$, the safety condition $\xi(\boldsymbol{\tau})$ for a trajectory $\boldsymbol{\tau}$ can be computed as

$$\xi(\boldsymbol{\tau}) = \int_{z_1, \ldots, z_m \geq 0} \mathcal{N}\left(\boldsymbol{\zeta} | \boldsymbol{\mu}_g(\boldsymbol{\tau}), \boldsymbol{\Sigma}_g(\boldsymbol{\tau})\right) \, dz_1, \ldots, z_m > 1 - \alpha . \tag{5}$$

In general, the computation of $\xi(\boldsymbol{\tau})$ is analytically intractable, and thus needs to rely on some approximation, such as Monte-Carlo sampling or expectation propagation [15].

### 3.3 The Algorithm

In the previous sections, we elaborated on the modeling of the predictive distribution and the safety condition for a given piecewise trajectory $\boldsymbol{\tau}$ in the input space. For efficiently choosing an optimal $\boldsymbol{\tau}$, the trajectory needs to be appropriately parametrized. The most straightforward possibility is to parametrize in the input space. We illustrate the trajectory parametrization in the following Example 2, using ramp parameterization.

**Example 2** (Consecutive ramps as piecewise trajectory). *A ramp can be parametrized with its start and end point. As the start point is the last point of the previous trajectory, the end point $\boldsymbol{\eta}$ is the only free quantity, and therefore a ramp can be parametrized as*

$$\boldsymbol{\tau}(\boldsymbol{\eta}) = (\boldsymbol{x}_1(\boldsymbol{\eta}), \ldots, \boldsymbol{x}_m(\boldsymbol{\eta}))$$
$$\text{with for } 1 \leq k \leq m : \ \boldsymbol{x}_k(\boldsymbol{\eta}) = \left( \boldsymbol{u}_0 + \frac{k}{m}(\boldsymbol{\eta} - \boldsymbol{u}_0), \ldots, \boldsymbol{u}_0 + \frac{k - d_2 + 1}{m}(\boldsymbol{\eta} - \boldsymbol{u}_0) \right) \tag{6}$$

*where $\boldsymbol{u}_0$ is the start point of the ramp. For $k - i \geq 0$, the manipulated input variable is on the currently planned trajectory, and for $k - i < 0$ it can be read from the list of already executed trajectories.*

Given a trajectory parametrization with its predictive distribution in Eq. (1) and safety condition in Eq. (5), the next piecewise trajectory $\boldsymbol{\tau}_{n+1}(\boldsymbol{\eta}^*)$ can be obtained by solving the following constrained optimization problem

$$\boldsymbol{\eta}^* \quad = \operatorname{argmax}_{\boldsymbol{\eta} \in \Pi} \ \mathcal{I}(\boldsymbol{\Sigma}(\boldsymbol{\eta})) \tag{7}$$
$$s.t. \ \xi(\boldsymbol{\eta}) > 1 - \alpha , \tag{8}$$

where $\boldsymbol{\eta}$ represents our trajectory parametrization, $\Pi$ is domain of the manipulated variable, and $\mathcal{I}$ an optimality criterion we will discuss later. As shown in Eq. (7), we employ the predictive variance $\boldsymbol{\Sigma}$ from Eq. (1) for the exploration, which is common in the active learning setting, especially in combination with a GP model [20, 14]. In contrast to previous work, due to the nature of the considered trajectory, we have a *covariance matrix* $\boldsymbol{\Sigma}$ instead of the variance value usually employed in the active learning and Bayesian optimization setting [23]. The covariance matrix is mapped by an optimality criterion $\mathcal{I}$ to a real number, as indicated by Eq. (7). Various optimality criteria can be used for $\mathcal{I}$, as discussed in the system identification literature [8]. For example, $\mathcal{I}$ can be the *determinant*, equivalent to maximizing the volume of the predictive confidence ellipsoid of the multi-normal distribution, the *trace*, equivalent to maximizing the average predictive variance, or the *maximal eigenvalue*, equivalent to maximizing the largest axis of the predictive confidence ellipsoid [8].

The constraint in Eq. (8) represents a probabilistic safety criterion, motivated by our probabilistic modeling approach for the safety. The probabilistic approach flexibly allows us to control the trade-off between exploration speed and safety consideration. For example, a 100% safe exploration would keep the algorithm from leaving the initial safe area and, hence, would not lead to an exploration of new areas. On the other hand, a 0% safe exploration will explore without any safe considerations which will result in many safety violations. This trade-off provides the users an additional degree of freedom, depending on how much they "trust" the behavior of their physical systems.

**Algorithm 1** Safe Active Learning for Time-Series Modeling

---
1: Input: Safety threshold $0 \leq \alpha \leq 1$
2: Initialization: Collect $n_0$ safe trajectories, i.e. $\mathcal{D}_0^{f,g} = \{\boldsymbol{\tau}_i, \boldsymbol{\rho}_i, \boldsymbol{\zeta}_i\}_{i=1}^n$ with $n = n_0$.
3: **for** $k = 1$ **to** $N$ **do**
4:     Update regression model approximating $f$ using $\mathcal{D}_{k-1}^f = \{\boldsymbol{\tau}_i, \boldsymbol{\rho}_i\}_{i=1}^n$, according to Eq. (1)
5:     Update safety model approximating $g$ using $\mathcal{D}_{k-1}^g = \{\boldsymbol{\tau}_i, \boldsymbol{\zeta}_i\}_{i=1}^n$, according to Eq. (4)
6:     Determine new piecewise trajectory $\boldsymbol{\tau}_{n+1}$, by optimizing $\boldsymbol{\eta}$ according to Eq. (7 and 8)
7:     Execute $\boldsymbol{\tau}_{n+1}$ on the physical system, while measuring $\boldsymbol{\rho}_{n+1}$ and $\boldsymbol{\zeta}_{n+1}$
8:     Include new trajectories into $\mathcal{D}_{k-1}^f$ and $\mathcal{D}_{k-1}^g$ with $n = n+1$.
9: **end for**
10: Update and return regression model and safety model

---

Algorithm 1 summarizes the basic steps of the proposed algorithm, which needs to be initialized by $n_0$ safe trajectories. In practice, the initial trajectories are located in a small, safe region chosen beforehand using prior knowledge. The incremental updates of the GP models for new data, i.e. steps 4 and 5 in Algorithm 1, can be efficiently performed, e.g. through rank-one updates [21]. The optimization problem in Eq. (7) can be solved using gradient-based optimization approaches, e.g. [4, 5]. In this paper, we employ the NX-structure in combination with the GP model for time-series modeling. However, this approach can also be extended to the general nonlinear auto-regressive exogenous case [3], i.e. a GP with NARX input structure $\boldsymbol{x}_k = (y_k, y_{k-1}, \ldots, y_{k-q}, u_k, u_{k-1}, \ldots, u_{k-d})$. In this case, for optimization and planning of the next piecewise trajectories, one can use the predictive mean of $p(\boldsymbol{\rho}|\boldsymbol{\tau}, \mathcal{D}_n^f)$ as surrogate for $y_k$. Note that the input excitation is still performed through the manipulated variable $u_k$ in the case of NARX.

## 4 Theoretical Results

In this section, we provide some results on the theoretical analysis of the proposed approach. First, we investigate the safety aspect of the algorithm. In Section 4.2, we provide a bound on the decay rate of the predictive variances for the case when the criterion $\mathcal{I}$ is a determinant, i.e. $\mathcal{I}(\boldsymbol{\Sigma}(\boldsymbol{\eta})) = \det(\boldsymbol{\Sigma}(\boldsymbol{\eta}))$. The proofs can be found in the Appendix.

### 4.1 Safe Exploration

To satisfy the safety requirements, it is necessary to bound the probability of failures during exploration. Theorem 1 provides an upper bound on the probabilities for unsafe trajectories.

**Theorem 1.** *Let us assume that we have recorded $n_0$ initial safe trajectories, and that their observations are enough to model $g$ well, in the sense that our GP quantifies the uncertainty of predictions for $g$ correctly, i.e. $P(\mu_g - \nu\sigma_g \leq z \leq \mu_g + \nu\sigma_g) = Erf(\nu/\sqrt{2})$ for all $\nu \geq 0$. Let $\delta \in [0, 1]$ be the desired failure probability when determining the next $N$ consecutive piecewise trajectories. Set $\alpha = \delta/N$ and let this $\alpha$ be the probability bound for a trajectory being unsafe (as in Eq. (5)). Then, the iterative exploration for the next $N$ trajectories is unsafe with probability at most $\delta$, i.e.*

$$P\left(\cup_{i=n_0+1}^{n_0+N} \left\{ g(\boldsymbol{x}_j^i) < 0 \text{ for some } 1 \leq j \leq m | \xi(\boldsymbol{\tau}_i) > 1 - \alpha \right\} \right) \leq \delta.$$

Theorem 1 supplies us with a useful rule of thumb to select $\alpha$ for sequentially determining the next $N$ trajectories.

### 4.2 Decay of Predictive Variance

The remainder of the analysis is to show that the proposed exploration scheme makes the predictive uncertainty $\boldsymbol{\Sigma}$ decrease as $n$ increases. In this paper we use the *determinant* of $\boldsymbol{\Sigma}$ as an exploration criterion, which has been shown to have a close relationship to the information gain [14, 23], defined as the mutual information $I$.

First, we point out that this relationship still holds true in case of trajectories as observations. Subsequently, we introduce the maximum information gain as an upper bound, which can further be

used to show the decrease of the predictive uncertainty. Lemma 1 clarifies the relationship between determinant and mutual information. Let us denote the predictive variance after recording $i-1$ trajectories as $\boldsymbol{\Sigma}_{i-1}(\boldsymbol{\tau}_i)$, $\boldsymbol{\Sigma}_0(\boldsymbol{\tau}_1) = \boldsymbol{k}^{**}(\boldsymbol{\tau}_1, \boldsymbol{\tau}_1)$, and set $\tilde{\boldsymbol{\rho}}_i = \left(f(\boldsymbol{x}_1^i), \ldots, f(\boldsymbol{x}_m^i)\right)$.

**Lemma 1.** *The mutual information* $I(\{\boldsymbol{\rho}_i\}_{i=1}^n; \{\tilde{\boldsymbol{\rho}}_i\}_{i=1}^n)$ *can be related to the predictive co-variances* $\boldsymbol{\Sigma}_{i-1}(\tau_i)$ *as follows*

$$I\left(\{\boldsymbol{\rho}_i\}_{i=1}^n; \{\tilde{\boldsymbol{\rho}}_i\}_{i=1}^n\right) = 1/2 \sum_{i=1}^n \log |\boldsymbol{I}_m + \sigma^{-2} \boldsymbol{\Sigma}_{i-1}(\boldsymbol{\tau}_i)|$$

Next, we introduce the maximum information gain after observing $n$ trajectories as $\gamma_n :=$ $\max_{\{\boldsymbol{\tau}_i\}_{i=1}^n \subset X^m} I(\{\boldsymbol{\rho}_i\}_{i=1}^n; \{\tilde{\boldsymbol{\rho}}_i\}_{i=1}^n)$, (see Srinivas et al. [23] for more details). The maximum information gain is the information which could be gathered when exploring the system in a non-iterative way, by optimally designing all trajectories simultaneously (which is in practice hard as it would require a solution of a high dimensional optimization problem, and would not allow us to incorporate safety information from observations during the experiment). According to Srinivas et al. ([23], Theorem 5) the maximum information gain satisfies $\gamma_n = O\left(log(n)^{d+1}\right)$, i.e. the maximum information grows slower than the number of additional trajectories. This will be crucial later on, but first we investigate the relation between the determinant of the covariance and $\gamma_n$. Using Lemma 1 the determinant of the covariance can be bounded, as given in Lemma 2.

**Lemma 2.** *After observing $n$ trajectories $\{\boldsymbol{\tau}_i\}_{i=1}^n$ (according to Eq. (7)), the determinant of the covariance is upper bounded by*

$$\frac{1}{n} \sum_{i=1}^n |\boldsymbol{\Sigma}_{i-1}(\boldsymbol{\tau}_i)| \leq C \frac{\gamma_n}{n},$$

*where $\boldsymbol{\Sigma}_{i-1}$ is the predictive variance computed using the previous $i-1$ trajectories, $\gamma_n$ is the maximum information gain, and $C = 2\sigma_f^{2m}/log(1 + \sigma^{-2m}\sigma_f^{2m})$ is a constant.*

The first step in proving Lemma 2 is to upper bound the predictive variance using the mutual information via Lemma 1. Subsequently, the mutual information is upper bounded by $\gamma_n$. Using Lemma 2 and Theorem 5 in [23], we can provide a decay rate on the average determinant of predictive variances.

**Theorem 2.** *Let $\{\tilde{\boldsymbol{\tau}}_i\}_{i=1}^n$ be $n$ arbitrary trajectories within a compact and convex domain $X$, and $k$ be a kernel function such that $k(\cdot, \cdot) \leq 1$. If $\boldsymbol{\Sigma}_{i-1}$ come from our exploration scheme Eq. (7) (i.e. without safety considerations), then we have*

$$\frac{1}{n} \sum_{i=1}^n |\boldsymbol{\Sigma}_{i-1}(\tilde{\boldsymbol{\tau}}_i)| = \mathcal{O}\left(\frac{\log(n)^{d+1}}{n}\right).$$

We sketch the proof here: as our algorithm (without safety considerations) always chooses the trajectory with the highest determinant (D criterion), the average determinant of an actively learned scheme is always higher than or equal to the average determinant of an arbitrary scheme. Therefore, $\frac{1}{n} \sum_{i=1}^n |\boldsymbol{\Sigma}_{i-1}(\tilde{\boldsymbol{\tau}}_i)| \leq \frac{1}{n} \sum_{i=1}^n |\boldsymbol{\Sigma}_{i-1}(\boldsymbol{\tau}_i)|$, which is $\mathcal{O}(log(n)^{d+1}/n)$ when employing Lemma 2 and the Theorem 5 of [23].

By Theorem 2, for any sequence of trajectories, the average of the determinants of their predictive covariances tends to zero. As the determinant corresponds to the volume of the confidence ellipsoid, we can conclude that the average volume of confidence ellipsoids tends to zero as well, indicating that on average, our predictions become precise. However, as the safety constraint in Eq. (8) changes at every iteration, we extend the statement of Theorem 2:

**Theorem 3.** *Let us assume that there exists a compact and convex domain $X$, and a kernel function $k$ such that $k(\cdot, \cdot) \leq 1$, that covers the whole area which is explorable (independent of whether it is safe or not). Then, the statement of Theorem 2 still holds for our Algorithm 1 with iteration-dependent safe areas $S_i$.*

Theorem 3 guarantees the decay of averaged determinants of covariances during safe exploration.

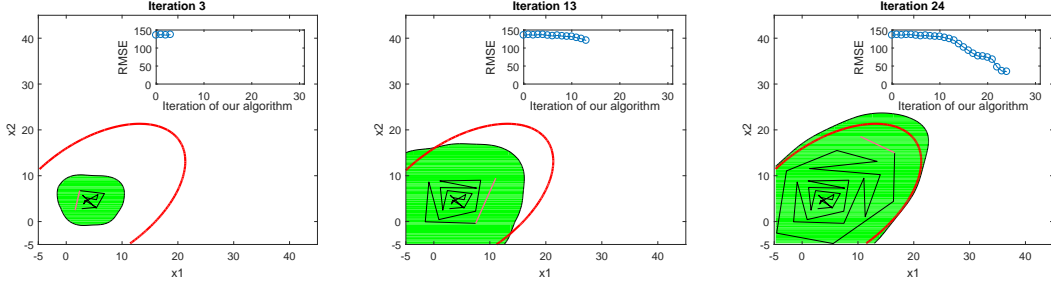

Figure 1: The columns show the progress of the approximation of $f$ (inlay) and the identified safety region (main figure) at different iterations. Each iteration corresponds to a consecutive planning of a new piecewise trajectory (here: 2D ramp). As shown by the results, the current estimation of the safe region (green area) gradually covers the actual safe area (red line), and the approximation error gradually decreases (as shown in the subfigures). An illustrative video showing all iterations can be found in the Appendix.

## 5 Evaluations

In section 5.1 we illustrate the proposed approach using synthetic models, comparing our safe active learning approach (SAL-NX) with random selection using safety constraints. Subsequently, we employ the approach to learn a dynamics model of a physical, high-pressure fluid system in Section 5.2. For simplicity we employ ramps for the piecewise trajectory parametrization, but other curve parameterizations could also be used instead, e.g. spline parameterization. The form of the input trajectory has an impact on the excitation of the system, as comprehensively studied in the field of system identification [17].

### 5.1 Simulated Experiments

**Experiment 1** In this experiment, a toy example is employed to illustrate the concept of input space exploration with piecewise trajectories and safe region detection. A function $f : \mathbb{R}^2 \to \mathbb{R}$, $f(\boldsymbol{x}) = (x^{(1)} - 2)^2 + (x^{(1)} - 2)(x^{(2)} - 2) + (x^{(2)} - 2)^2$ with $\boldsymbol{x} = (x^{(1)}, x^{(2)})$ is used as the ground-truth. An observation is given by $y = f(\boldsymbol{x}) + \epsilon$ with $\epsilon \sim \mathcal{N}(0, 1)$. The safe region is characterized by $g : \mathbb{R}^2 \to \mathbb{R}$ with $g(\boldsymbol{x}) = (x^{(1)} - 5)^2 + (x^{(1)} - 5)(x^{(2)} - 5) + (x^{(2)} - 5)^2$. The safety indicator $z$ is given as $z = -0.005 \cdot g(\boldsymbol{x}) + 1 + \varsigma$ with $\varsigma \sim \mathcal{N}(0, 1)$. It is considered to be safe for $z > 0$, otherwise unsafe.

We proceed as shown in Algorithm 1, where the piecewise trajectories are parametrized as 2D-ramps with 5 discretization points (i.e. $m = 5$, see Example 2). We start with 10 initial safe trajectories and consecutively determine new piecewise trajectories in the input space $X$, while also collecting outputs $y$ and computing safety indicator values $z$. As the exploration progresses, the approximation of $f$ and $g$ becomes more and more accurate, as shown in Fig. 1. The current estimation of the safe region (green area) gradually covers the actual safe area, and the approximation error gradually decreases (as shown in the subfigures). An illustrative video showing all iterations can be found in the Appendix.

**Experiment 2** In this experiment, we learn a time-series model given as a GP with NX-structure. We have two manipulated variables $u_k^{(1)}$ and $u_k^{(2)}$ at time $k$. The NX-structure is determined to be $\boldsymbol{x}_k = (u_k^{(1)}, u_k^{(2)}, u_{k-1}^{(1)}, u_{k-1}^{(2)})$, an input space with $d = 4$. The ground-truth models of $f$ and $g$ are provided in the Appendix. The piecewise trajectory is again parametrized as 4D-ramps with $m = 5$. We initialize the models using 10 collected piecewise ramps in a safe area, and start exploring in the input space. For a fair comparison, we benchmark the proposed algorithm against a *random selection with safe constraints* of next piecewise trajectories. Instead of optimizing the ramp parameter $\boldsymbol{\eta}$ as shown in Eq. (7) and (8), we randomly select $\boldsymbol{\eta}$ and pick the first one which fulfills the safety constraint $\xi(\boldsymbol{\eta}) > 1 - \alpha$. Fig. 2 shows the results of the comparison of the proposed approach (SAL-NX) with random selection.

The results in Fig. 2 show that SAL-NX continuously improves the model approximation (shown as RMSE) and provides fast coverage of safe regions. The models for $f$ and $g$ are updated after every iteration by including new sample points. The hyperparameters can be estimated beforehand

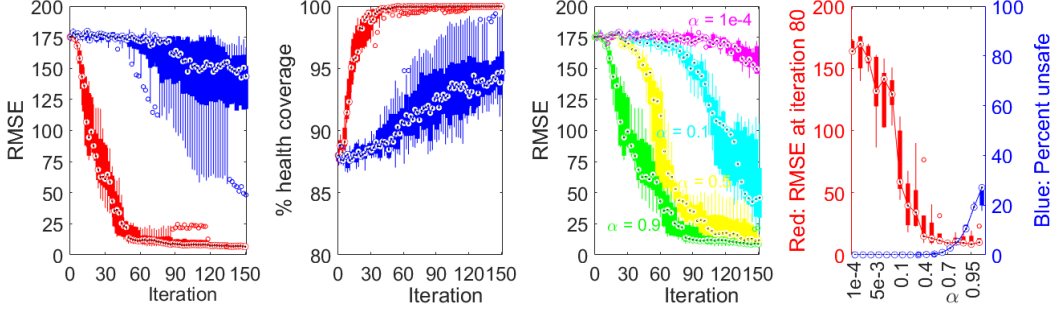

Figure 2: The **first two** pictures from the left show the comparison of the SAL-NX (red line) with random selection (blue line). SAL-NX yields faster convergence in model approximation (left picture) and coverage of safe regions (right picture), while having less variance and outliers (indicated as small circles). The **last two** pictures show the impact of the safety threshold $\alpha$. The left picture shows the RMSE of SAL-NX for 4 different values of $\alpha$. The right picture shows the model approximation error as RMSE (red line) and percentage of *unsafe* trajectories (blue line) as a function of $\alpha$. All pictures show boxplots over 5 repetitions.

or updated after every iteration. For the required number of initial trajectories, we refer to the lower bound as given in [20]. For computing the safety condition $\xi(\boldsymbol{\tau})$ from Eq. (5), we employ Monte-Carlo sampling. Our experiments are performed on a desktop computer. The algorithm is sufficiently fast for real-time applications.

In this experiment, we also compare our exploration approach with the one proposed in [6], however, *without* safety requirements in order to cope with the setting from [6]. We adapt their criterion based on the Fisher information for our GP model by employing the GP mean function. Additionally, we also compare the decrease in RMSE of the Fisher information based criterion to the decrease in RMSE of our algorithm. The results can be found in the Appendix 7.4 and show a competitive performance of our approach.

## 5.2 Learning a Surrogate Model of the High-Pressure Fluid System

**The Use Case**  As a realistic technical use case, we employ the approach to actively learn a surrogate model of a high-pressure fluid injection system, as shown in Figure 3. Such systems are widely used in industry, e.g. in the automotive domain for injection of fuel into the combustion engine [25].

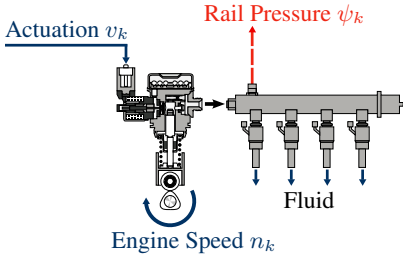

Figure 3: High-pressure fluid injection system with controllable inputs $v_k$, $n_k$ and measured output $\psi_k$ (picture taken from [25]).

The physical injection system is controlled by an actuation signal $v_k$ and the speed of an external engine $n_k$, for every time step $k$. The goal is to obtain a surrogate model predicting the rail pressure $\psi_k$, which determines the amount of fluid coming out of the outlets. Due to the nature of the fluid and the mechanical components, the dynamics of the whole system are nonlinear, and thus model learning is an appropriate alternative compared to analytical models. However, generating the data for learning a time series surrogate model by varying the actuation signal and engine speed is not simple, as an inappropriate combination of them would result in hazardously high rail pressures, damaging the physical system.

**Learning Time-Series Surrogate Models**  Due to the safety requirements and the fact that the safety boundary is not known beforehand, our safe active learning approach is very appropriate for approximating the dynamics model. The employed NX-structure is chosen to be $\boldsymbol{x}_k = (n_k, n_{k-1}, n_{k-2}, n_{k-3}, v_k, v_{k-1}, v_{k-3})$. We again parameterize with piecewise ramps in this 7D input space. The safety indicator value $z$ is computed as shown in Example 1, with $\psi_{max} = 18$ MPa being the maximally allowed rail pressure. It should be noted that here $z$ is computed as a function of the target output $\psi$, in constrast to the experiments in Section 5.1, where $z$ is a function of the input. We initialize the model with 25 trajectories sampled around a safe point chosen by a domain expert. Subsequently, we start exploring the input space dynamically, considering both the safety constraint

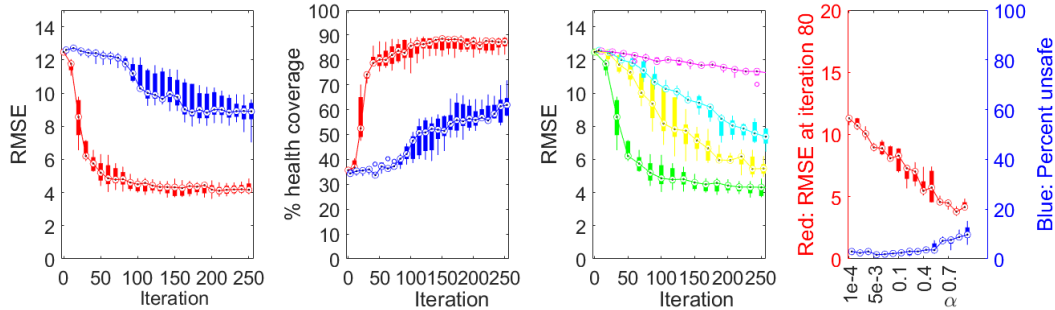

Figure 4: The **first two** pictures from the left show the comparison of the SAL-NX (red line) with random selection with safe constraints (blue line), with respect to model approximation and coverage of safe regions. Here, $\alpha = 0.5$ and 250 trajectories are planned. The **last two** pictures show the impact of the safety threshold $\alpha$ on the approximation error, and failures during exploration. The results are displayed as a boxplot over 5 repetitions.

and the model information gain, while measuring the actuation and speed signals as input and the rail pressure as output.

Figure 4 shows the results after exploring the input space with 250 consecutive ramp trajectories, each consisting of $m = 5$ discretization points. We update the hyperparameters after every iteration. We compare our SAL-NX approach with the random selection, as described in the previous experiment. The figure also shows the impact of varying the threshold value $\alpha$ on both the model approximation error and the percenteage of selected unsafe trajectories. In practice, the execution of the trajectories on the physical system is interrupted, when the system notices a violation of the maximal pressure $\psi_{max}$. The selected piecewise trajectory is then indicated as unsafe. For the evaluation of the coverage (second picture from the left), the "ground-truth" safe region is estimated beforehand with an extensive procedure.

# 6   Conclusions

In this paper we present an approach for active learning of a time-series model, given as a GP model with NX-structure. In this setting, the exploration is performed while taking safety requirements into account. For the successful application of the algorithm, it is crucial that the system can be actively controlled by a set of inputs and a safety signal can be observed during the system's operation. The proposed approach is evaluated on toy examples, as well as on a realistic technical use case. The results show that this approach is appropriate for real-world applications, especially, in the industrial setting, where safety is a key requirement during operation.

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
