[Supplementary Material · Safe Active Learning for Time-Series Modeling with Gaussian Processes Appendix.pdf]

# Safe Active Learning for Time-Series Modeling with Gaussian Processes

Christoph Zimmer[(1)], Mona Meister[(1)] and Duy Nguyen-Tuong[(1)]

[1] Bosch Center for Artificial Intelligence, Robert Bosch GmbH, Renningen, Germany

## 7 Appendix

This Appendix contains the proof of the safety theorem, the proofs of the lemmas and theorems on the variance decay as well as some details on the numerical evaluation section.

### 7.1 Proof for the Safety Theorem

In the following, we prove Theorem 1, providing a safety guarantee for algorithm 1.

*Proof of Theorem 1.* According to the constraint in Eq. (5), each trajectory $\boldsymbol{\tau}_i$, $i = n_0 + 1, \ldots, n_0 + N$ is unsafe with probability at most $\alpha$ (if the GP quantifies the uncertainty of predictions for $g$ correctly). Thus, a trajectory $\boldsymbol{\tau}_i$ is critical (it is unsafe but considered as safe), if $\xi(\boldsymbol{\tau}_i) > 1 - \alpha$ and $g(\boldsymbol{x}_j^i) < 0$ for at least one $1 \le j \le m$. Using the union bound, we have

$$P\left(\cup_{i=n_0+1}^{n_0+N} \left\{ g(\boldsymbol{x}_j^i) < 0 \text{ for at least one } 1 \le j \le m | \xi(\boldsymbol{\tau}_i) > 1 - \alpha \right\}\right)$$

$$\le \sum_{i=n_0+1}^{n_0+N} P\left(\left\{ g(\boldsymbol{x}_j^i) < 0 \text{ for at least one } 1 \le j \le m | \xi(\boldsymbol{\tau}_i) > 1 - \alpha \right\}\right)$$

$$\le \sum_{i=n_0+1}^{n_0+N} \alpha = N\,\alpha = N\,\delta/N = \delta \ .$$

Note that $\xi(\boldsymbol{\tau}_i)$ is a random variable because $\boldsymbol{\rho}$ is a random variable, and therefore $\boldsymbol{\mu}_g$ and $\boldsymbol{\Sigma}_g$ as well, depending on the previously recorded values for $\boldsymbol{\rho}_j$, $1 \le j \le i - 1$. Additionally, $g(\boldsymbol{x}_j^i)$ is a random variable as well (even though $g$ is deterministic) because the choice of trajectories (and therefore points $\boldsymbol{x}_j^i$) depends on the optimization problem (Eq. 7-8), the solution of which depends on $g$, which itself depends on $\boldsymbol{\rho}$. ∎

### 7.2 Proofs on Reduction of Uncertainty

Next, we will prove the lemmas and theorems on the variance decay from section 4.2. The proofs for Lemma 1, 3, and 4 are motivated by Lemma 5.3 and 5.4 of [23] but extended to the situation of trajectories.

First, we prove Lemma 1 to relate the information gain $I$ to the covariance $\Sigma$.

*Proof of Lemma 1.* The mutual information can be expressed in terms of entropy as

$$I\left(\{\boldsymbol{\rho}_i\}_{i=1}^n; \{\tilde{\boldsymbol{\rho}}_i\}_{i=1}^n\right) = H\left(\{\boldsymbol{\rho}_i\}_{i=1}^n\right) - H\left(\{\boldsymbol{\rho}_i\}_{i=1}^n | \{\tilde{\boldsymbol{\rho}}_i\}_{i=1}^n\right)$$

As $\{\boldsymbol{\rho}_i\}_{i=1}^n | \{\tilde{\boldsymbol{\rho}}_i\}_{i=1}^n$ is Gaussian with variance $\sigma^2 I_{nm}$, it holds that $H(\{\boldsymbol{\rho}_i\}_{i=1}^n | \{\tilde{\boldsymbol{\rho}}_i\}_{i=1}^n) = 1/2 \log |2\pi\, e\sigma^2 I_{nm}| = nm/2 \log(2\pi e\sigma^2)$. Furthermore, using entropy rules, we obtain

$$H\left(\{\boldsymbol{\rho}_i\}_{i=1}^n\right) = H\left(\{\boldsymbol{\rho}_i\}_{i=1}^{n-1}\right) + H\left(\boldsymbol{\rho}_n | \{\boldsymbol{\rho}_i\}_{i=1}^{n-1}\right)$$

and

$$H(\boldsymbol{\rho}_n | \{\boldsymbol{\rho}_i\}_{i=1}^{n-1}) = 1/2 \log\left(|2\pi\, e(\sigma^2 \boldsymbol{I}_m + \boldsymbol{\Sigma}_{n-1}(\tau_n))|\right)$$

$$= 1/2 \log\left(|2\pi\, e\, \sigma^2(\boldsymbol{I}_m + \sigma^{-2}\boldsymbol{\Sigma}_{n-1}(\tau_n))|\right)$$

$$= 1/2 \log\left((2\pi\, e\, \sigma^2)^m\right) + 1/2 \log\left(|(\boldsymbol{I}_m + \sigma^{-2}\boldsymbol{\Sigma}_{n-1}(\tau_n))|\right) \ .$$

Alltogether, this leads to

$$
\begin{aligned}
I\left(\{\boldsymbol{\rho}_i\}_{i=1}^n; \{\tilde{\boldsymbol{\rho}}_i\}_{i=1}^n\right) &\\
= H\left(\{\boldsymbol{\rho}_i\}_{i=1}^n\right) &- H\left(\{\boldsymbol{\rho}_i\}_{i=1}^n | \{\tilde{\boldsymbol{\rho}}_i\}_{i=1}^n\right) \\
= \sum_{j=1}^n H(\boldsymbol{\rho}_i | \{\boldsymbol{\rho}_i\}_{i=1}^{j-1}) &- nm/2 \log(2\pi e \sigma^2) \\
= \sum_{j=1}^n \left(1/2 \log\left((2\pi\, e\, \sigma^2)^m\right)\right) &+ \sum_{j=1}^n \left(1/2\log\left(|(\boldsymbol{I}_m + \sigma^{-2}\boldsymbol{\Sigma}_{j-1}(\tau_j))|\right)\right) - nm/2\log(2\pi e \sigma^2) \\
= 1/2 \sum_{i=1}^n \log |\boldsymbol{I}_m + \sigma^{-2}\boldsymbol{\Sigma}_{i-1}(\boldsymbol{\tau}_i)| \; . &
\end{aligned}
$$

∎

Lemma 1 relates the information $I$ to the covariance $\Sigma$, but not yet to the determinant of the covariance $|\Sigma|$. To link the determinant of the covariance $|\Sigma|$ to the mutual information in Lemma 4, we first state the auxilliary Lemma 3.

**Lemma 3.** *For* $i = 1, \dots, n$,

$$
|\boldsymbol{\Sigma}_{i-1}(\tau_i)| \le C_1 \log(1 + |\sigma^{-2}\boldsymbol{\Sigma}_{i-1}(\tau_i)|)
$$

*is satisfied, where* $C_1 = \frac{\sigma_f^{2m}}{\log(1+\sigma^{-2m}\sigma_f^{2m})}$.

*Proof.* Let $e_1, \dots, e_m$ be the eigenvalues of $\boldsymbol{\Sigma}_{i-1}(\tau_i)$. Then it holds that $|\boldsymbol{\Sigma}_{i-1}(\tau_i)| = \prod_{i=1}^m e_i$ and furthermore that $\text{trace}(\boldsymbol{\Sigma}_{i-1}(\tau_i)) = \sum_{i=1}^m e_i$. Keeping in mind that the geometric mean is less than or equal to the arithmetic mean, we have

$$
\sqrt[m]{|\boldsymbol{\Sigma}_{i-1}(\tau_i)|} \le \frac{\text{trace}(\boldsymbol{\Sigma}_{i-1}(\tau_i))}{m} \; .
$$

As the trace of $\boldsymbol{\Sigma}$ contains only predictive variances (and no correlation terms), we can upper bound them by

$$
\frac{\text{trace}(\boldsymbol{\Sigma}_{i-1}(\tau_i))}{m} = \frac{\sum_{j=1}^m \sigma_{i-1}(x_j^i)}{m} \le \sigma_f^2 \; ,
$$

where $x_j^i$, $j = 1, \dots, m$, $i = 1, \dots, n$, are the $m$ points of the $i$-th trajectory. This results in the upper bound

$$
|\boldsymbol{\Sigma}_{i-1}(\tau_i)| \le \sigma_f^{2 \cdot m} \; .
$$

Now, it holds that $a := \sigma^{-2m}|\boldsymbol{\Sigma}_{i-1}(\tau_i)| \le \sigma^{-2m}\sigma_f^{2m} =: b$. Furthermore, we have

$$
\begin{aligned}
\log(1 + a) = \log\left(\frac{a}{b}(1 + b) + (1 - \frac{a}{b}) \cdot 1\right) & \\
\ge \frac{a}{b}\log(1 + b) + (1 - \frac{a}{b})\log(1) & \quad (9) \\
= \frac{a}{b}\log(1 + b) \; , &
\end{aligned}
$$

where the second step follows with the concavity of the log function. With this in hand, we can now conclude that

$$
\begin{aligned}
|\boldsymbol{\Sigma}_{i-1}(\tau_i)| &= \sigma^{2m}\,\sigma^{-2m}\,|\boldsymbol{\Sigma}_{i-1}(\tau_i)| \\
&\le \sigma^{2m} \frac{\sigma^{-2m}\sigma_f^{2m}}{\log(1 + \sigma^{-2m}\sigma_f^{2m})} \log\left(1 + \sigma^{-2m}|\boldsymbol{\Sigma}_{i-1}(\tau_i)|\right) \\
&= \sigma^{2m} \frac{\sigma^{-2m}\sigma_f^{2m}}{\log(1 + \sigma^{-2m}\sigma_f^{2m})} \log\left(1 + |\sigma^{-2}\boldsymbol{\Sigma}_{i-1}(\tau_i)|\right) \\
&= C_1 \log\left(1 + |\sigma^{-2}\boldsymbol{\Sigma}_{i-1}(\tau_i)|\right) \; ,
\end{aligned}
\qquad (10)
$$

where $C_1 := \frac{\sigma_f^{2m}}{\log(1+\sigma^{-2m}\sigma_f^{2m})}$. ∎

**Lemma 4.** *The sum of the determinants of variances is upper bounded by the mutual information, i.e.*

$$
\sum_{i=1}^n |\boldsymbol{\Sigma}_{i-1}(\tau_i)| \le C\, I\left((\boldsymbol{\rho})_{i=1}^n; \{\tilde{\boldsymbol{\rho}}\}_{i=1}^n\right) .
$$

*Proof.* For $C := 2C_1$, it holds

$$\sum_{i=1}^{n} |\mathbf{\Sigma}_{i-1}(\tau_i)| \leq \sum_{i=1}^{n} C_1 \log \left(1 + |\sigma^{-2}\mathbf{\Sigma}_{i-1}(\tau_i)|\right)$$

$$= \sum_{i=1}^{n} C_1 \log \left(|\mathbf{I}| + |\sigma^{-2}\mathbf{\Sigma}_{i-1}(\tau_i)|\right)$$

$$\leq \sum_{i=1}^{n} C_1 \log \left(|\mathbf{I} + \sigma^{-2}\mathbf{\Sigma}_{i-1}(\tau_i)|\right)$$

$$= C \, I\left((\boldsymbol{\rho})_{i=1}^{n}; \{\tilde{\boldsymbol{\rho}}\}_{i=1}^{n}\right).$$

Here, the first step holds due to Lemma 3, the third as $|\mathbf{I}| + |\mathbf{A}| \leq |\mathbf{I} + \mathbf{A}|$ for a positive semidefinite matrix $\mathbf{A}$ as $|\mathbf{I}| + |\mathbf{A}| = \prod_{i=1}^{m} 1 + \prod_{i=1}^{m} a_i \leq \prod_{i=1}^{m}(1 + a_i) = |\mathbf{I} + \mathbf{A}|$ with $a_i$ being eigenvalues of $\mathbf{A}$ and $a_i \geq 0$ as $\mathbf{A}$ positive semidefinite. The last step holds according to Lemma 1. Note that $(\mathbf{A} + \mathbf{I})\boldsymbol{v_i} = \mathbf{A}\boldsymbol{v_i} + \mathbf{I}\boldsymbol{v_i} = a_i \cdot \boldsymbol{v_i} + \boldsymbol{v_i} = (a_i + 1)\boldsymbol{v_i}$ for $\boldsymbol{v_i}$ being an eigenvector for eigenvalue $a_i$ shows that $a_i + 1$ is indeed an eigenvalue of $\mathbf{A} + \mathbf{I}$. ∎

As we have now established a link between the determinant of the covariance – our criterion – and the mutual information $I$, we can upper bound the average of the determinant of the covariances and prove Lemma 2.

*Proof of Lemma 2.* According to Lemma 4 it holds

$$\sum_{i=1}^{n} |\mathbf{\Sigma}_{i-1}(\tau_i)| \leq C \, I\left(\{\boldsymbol{\rho}\}_{i=1}^{n}; \{\tilde{\boldsymbol{\rho}}\}_{i=1}^{n}\right).$$

Dividing by $n$ and upper bounding $I$ by $\gamma_n$ yields the desired results. ∎

This statement holds for trajectories from a run of our algorithm without safety. To ensure a good model, we are more interested in the behavior for arbirtrary trajectories, not only those that were chosen by our algorithm. Using a result from Srinivas [23], Theorem 2 extends the statement to an asymptotic behavior of the covariances of arbitrary trajectories.

*Proof of Theorem 2.* As our exploration scheme in Eq. (7) always chooses the trajectory with the highest determinant (D criterion), the average determinant of an actively learned scheme is always greater than or equal to the average determinant of an arbitrary scheme. Therefore:

$$\frac{1}{n} \sum_{i=1}^{n} |\mathbf{\Sigma}_{i-1}(\tilde{\tau}_i)| \leq \frac{1}{n} \sum_{i=1}^{n} |\mathbf{\Sigma}_{i-1}(\tau_i)|$$

which is $O\left(\frac{\log(n)^{d+1}}{n}\right)$ according to Lemma 2 and Theorem 5 of [23]. ∎

By Theorem 2, the average of the determinants of the covariance matrices tends to zero. However, it is phrased for a domain $X$ which is invariant over the iteration of our algorithm. This means it does not yet take safety constraints into account. To this end, Theorem 3 extends Theorem 2 such that it still holds for safe exploration.

*Proof of Theorem 3.* As it holds that $S_i \subseteq X$ for all $i$, it follows that

$$\max_{\tilde{\boldsymbol{\tau}}_i \in S_i^m} |\mathbf{\Sigma}_{i-1}(\tilde{\boldsymbol{\tau}}_i)| \leq \max_{\tilde{\boldsymbol{\tau}}_i \in X^m} |\mathbf{\Sigma}_{i-1}(\tilde{\boldsymbol{\tau}}_i)|$$

for $1 \leq i \leq n$. The remainder of the proof is analogous to the proof of Theorem 2. ∎

### 7.3 Numerical Evaluation Details

For Experiment 2 in Section 5.1 the ground truth regression function $f$ depends on the current state $u_k = \left(u_k^{(1)}, u_k^{(2)}\right)$ and the previous state $u_{k-1} = \left(u_{k-1}^{(1)}, u_{k-1}^{(2)}\right)$. Using $\boldsymbol{x}_k = (u_k, u_{k-1})$, the ground truth function is given by

$$f(x_k) = \tilde{f}(u_k) + \frac{\tilde{f}(u_k) - \tilde{f}\left(\left(u_{k-1}^{(1)}, u_k^{(2)}\right)\right)}{u_k^{(1)} - u_{k-1}^{(1)}} + \frac{\tilde{f}(u_k) - \tilde{f}\left(\left(u_k^{(1)}, u_{k-1}^{(2)}\right)\right)}{u_k^{(2)} - u_{k-1}^{(2)}} \qquad (11)$$

with a base function $\tilde{f} : [-5, 45]^2 \subset \mathbb{R}^2 \to \mathbb{R}$, where

$$\tilde{f}(u) = (u^{(1)} - 2)^2 - (u^{(1)} - 2)(u^{(2)} - 2) + (u^{(2)} - 2)^2 .$$

The observations are drawn from the model $y = f(\boldsymbol{x}) + \epsilon$ with noise $\epsilon \sim \mathcal{N}(0, 1)$.

The ground truth safety is constructed similarly with a base function $\tilde{g} : [-5, 45]^2 \subset \mathbb{R}^2 \to \mathbb{R}$ given by

$$\tilde{g}(u) = (u^{(1)} - 5)^2 - (u^{(1)} - 5)(u^{(2)} - 5) + (u^{(2)} - 5)^2$$

and a history dependent ground truth $g$

$$g(x_k) = \tilde{g}(u_k) - \left| \frac{\tilde{g}(u_k) - \tilde{g}\left((u_{k-1}^{(1)}, u_k^{(2)})\right)}{u_k^{(1)} - u_{k-1}^{(1)}} \right| - \left| \frac{\tilde{g}(u_k) - \tilde{g}\left((u_k^{(1)}, u_{k-1}^{(2)})\right)}{u_k^{(2)} - u_{k-1}^{(2)}} \right| .$$

The safety observations are drawn from the model $z = -0.005 * g(\boldsymbol{x}) + 1 + \epsilon_g$ with normally distributed noise $\epsilon_g$ with zero mean and standard deviation 0.01.

## 7.4 Benchmarking without safety consideration

We have benchmarked our newly proposed SAL-NX algorithm against a random selection approach and shown the superiority of our algorithm with respect to decreasing RMSE and speed of learning the safe area (Figures 2 and 4). Additionally, one might also think of a comparison to a more sophisticated approach than random selection. However, we are not aware of any algorithm that tackels all three challenges: active learning, safe learning and learning of dynamics models. There is an approach that is designed for dynamics model learning based on work of [6] and [7]. We reimplemented their approach and code as follows in order to allow for the comparison on model learning without taking safety considerations into account as their algorithm does not consider safety aspects.

The approach of [6] and [7] is based on building a Fisher information matrix. Intuitively speaking, the Fisher information matrix measures the sensitivity of the model response with respect to parameters – in our case the endpoints of the trajectories. By choosing a point with high sensitivity, one can place measurements to locations where they provide much information on the parameter. The Fisher Information matrix is given by

$$I_{\text{FIM}}(\boldsymbol{\eta}) = (I_{\text{FIM}}(\boldsymbol{\eta}))_{j=1,k=1}^{m,m}$$

with entries

$$(I_{\text{FIM}}(\boldsymbol{\eta}))_{j,k} = \frac{\partial \boldsymbol{\mu}(\boldsymbol{\tau}(\boldsymbol{\eta})^T)}{\partial \eta^{(j)}} \frac{\partial \boldsymbol{\mu}(\boldsymbol{\tau}(\boldsymbol{\eta}))}{\partial \eta^{(k)}} + \sum_{i=1}^{n-1} \frac{\partial \boldsymbol{\mu}(\boldsymbol{\tau}(\boldsymbol{\eta}_i))^T}{\partial \eta^{(j)}} \frac{\partial \boldsymbol{\mu}(\boldsymbol{\tau}(\boldsymbol{\eta}_i))}{\partial \eta^{(k)}}$$

where $\boldsymbol{\mu}(\boldsymbol{\tau}(\boldsymbol{\eta}_i))$ is the mean vector of the $i$-th trajectory with endpoint $\boldsymbol{\eta}_i$. Similarly as in equation (7), we choose the next trajectory by optimizing the information gain - now measured by the Fisher Information matrix

$$\boldsymbol{\eta}^* = \text{argmax}_{\boldsymbol{\eta} \in \Pi} \ \mathcal{I}\left(I_{\text{FIM}}(\boldsymbol{\eta})\right). \tag{12}$$

Figure 5 (left panel) shows a comparison similar to Figure 2 (left panel) between our proposed SAL-NX algorithm without safety consideration and the approach of [6] and [7] based on the Fisher information matrix. This first comparison does not look promising for the Fisher information matrix. The main difference of SAL-NX without safety consideration and the Fisher Information matrix is that SAL-NX reduces the predictive uncertainty while the Fisher information focuses on the slope of the GP mean prediction.

Figure 5 (right panel) shows how the safe area is learned. Note, that even though there is no safety restriction (as given by Eq. (8)) for the algorithm in exploring the input space, safety measurements $z$ are still recorded and, therefore, a safety model can be learned. Our SAL-NX approach seems to show competitive performance in learning the safety model as well.

None of both approaches without safety considerations leads to safe learning as in both approaches the percentage of safety violations is high: more than $40\%$ in average over 5 repetitions.

Figure 5: Right panel: As figure 2 but without safety consideration. Blue color representing the approach based on the Fisher information matrix, red color still representing our SAL-NX approach (now without safety consideration).

# 8   Appendix - Files

**Our SAL-NX algorithm safely learning a model – video.** File: SAL-NX-in-action.avi.
Upper panel: RMSE; lower panel: red - boundary between safe and unsafe area, green - learned as safe area, black - already executed trajectory, purple - new trajectory piece currently planned.

# 9   Code

The code will be released on GitHub.