[Reviews · NeurIPS 2018]

Reviewer 1



This paper proposes safe active learning algorithm for learning time-series models with piecewise trajectory sections. The basic model for the trajectory section is Gaussian Process, and the safety criterion is modeled as probability of certain safe value, which is modeled as a continuous function of inputs, being in the right region is greater than some threshold. The active learner explores the input space by generating trajectories that maximizes certain criterion in system identification. The paper also provides theoretical analysis in the safety perspective, as well as in the predictive variance reduction perspective. In the evaluation, besides simulated examples on predefined continuous functions and low dimensional models, this paper provides a specific use case for learning surrogate model of the high-pressure fluid system. Significance The active learning here is not based on pool of data or stream of data. This work is more of active generation of the piecewise input trajectory in order to learn system property models while receiving output from the physical system and keeping the generation in the safety region. Safe exploration in the same flavor I believe has been studied before in regression when there is some hard threshold. This paper generally makes good contribution to safe active learning in system identification. My concern though is that this work is not general enough for time series data. It mainly focuses on learning system models for physical systems. We need to have a good model for piecewise trajectory generation, we need to have continuous safety value function, and we need to have efficient optimization method for maximizing the optimality criteria when exploring. Therefore, my general suggestion here is to summarize all the conditions clearly in the paper for efficiently applying the method in practice. Quality I have several questions which I wish could be answered to help me better judge the quality of the paper: 1. There is this constrained optimization problem that needed to be solved for every iteration of exploration. And there is several different criteria mentioned. But I did not find anything about how to solve the problem efficiently. Also, in evaluation, example 2 does not solve the optimization problem while in the use case is not clearly mentioned what criteria is used there. 2. Instead of using some hard threshold as safety criterion, the probability based safety criterion in this paper seems just a little softer. What is the consideration here? 3. There are actually many small hyperparameters needed to be set beforehand for this method. For example, the length of the initialization trajectories and the timestep in each trajectory section. There is no discussion about these in the paper. Both 10 and 25 are used in different experiments. I want to know what is the general principle of choosing them. Originality This work reminds me of a previous thread of work on safe exploration based also on GP, but not on time series (like Safe Exploration for Optimization with Gaussian Processes), which is not cited in this paper. I think more discussion in related work may be needed. Clarity The paper is generally well-written with clear presentation and examples.

Reviewer 2



The submission proposes a Gaussian-Process based method for Safe Active Learning on time series. For Active Learning, the authors generalize the known GP-based techniques to estimate potential decrease in GP vairance (the full covariance matrix is taken into account). The novelty of the submission is claimed to be in that safety is taken into account. Rather than generating unconstrained time series to be tested in the real system, the approach also builds another GP model for safety of the system at every moment (assuming the data for safety is provided as well, for example from known physical properties of the system), and generating the times series under the constraints. Theorem providing probabilistic guarantees on the safety is proven. Another theorem proving the reduction in system variance is presented. Experiments with artificial data are performed, comparing random selection with the proposed method. Experiments with real fluid dynamics system are performed demonstrating the benefits of the proposed approach. The submission is well written, clear, interesting, and grounded. Theoretical analysis is well-justified and thorough, experimental analysis is mostly convincing. Some notes to the authors: * Please provide other baselines for the experiments than Random. If none of them work with safety constraints, demonstrate that the currently available techniques significantely break the safety guarantees. * Theorem 1 proves probabilistic guarantees. However, I can imagine for the real systems, it might be unacceptable to have any even rather small probbability of failure (e.g. the required bound is extremely tight). How does/can the proposed approach work for hard constraints on safety? ---- * On line 20, you mention that the data "has to be generated". There are some Active Learning methods for regression where there is no need to generate new data (e.g. referenced Guestrin et.al. paper), is that still the case for time series? Please refer to other time series-related work * You mention that there is work on safe exploration in RL. How novel is the current approach in comparison with those? Can this approach be used in RL? * On lines 73-81, it's really hard to distinguish $\tilde d$ and $\bar d$, please change the notation * On lines 150-153, there are references to various methods of working with I. Some discussion on whether the method selected for the submission is better/worse would be good to have. * In Theorem 1, what does "correctly" mean? * In Figure 2, right, it might be better to use a different axis for the blue line. * Please provide time for computation for the experiments and the method * It is hard to reproduce the experiments without code for the method and data. Please provide these with the final submission. ---- I have considered authors' feedback, and thank them for the explanations. My opinion about the acceptance remains the same

Reviewer 3



This paper introduces a a new, Gaussian-Process-based, model for Time-Series Modelling which takes safety regiouns into account. After an in-depth theoretical study, the bounds and "rule-of-thumbs" for setting the safety threshold are discussed. Finally, experiments on synthetic data and a Surrogate Model of the High-Pressure Fluid System are performed. Impressively, the method is better than random choice and the impact of the safety threshold is rather big. However, the paper has several flaws. The approach is interesting and sounds novel to me, but it is not compared with other active learning algorithms. Especially in the experiments typical methods could have been taken into account. You could, e.g., have compared to [3] in the limited environment. Unfortunately, the paper suffers from many linguistic mistakes (word choice and grammar). It is wise to proof-read it if it gets accepted.